# Opportunistic Multipath Routing in Long-Hop Wireless Sensor Networks

**DOI:** 10.3390/s19194072

**Published:** 2019-09-20

**Authors:** Sangdae Kim, Beom-Su Kim, Kyong Hoon Kim, Ki-Il Kim

**Affiliations:** 1Department of Informatics, Gyeongsang National University, Jinju 52828, Korea; sdkim.cse@gmail.com (S.K.); khkim@gnu.ac.kr (K.H.K.); 2Department of Computer Science and Engineering, Chungnam National University, Daejeon 34134, Korea; bumsou10@cnu.ac.kr

**Keywords:** long-hop wireless sensor networks (WSNs), multipath routing, opportunistic routing, reliability, energy efficiency

## Abstract

To improve the packet delivery ratio in wireless sensor networks, many approaches such as multipath, opportunistic, and learning-based routing protocols have been proposed. However, the performance of the existing protocols are degraded under long-hop wireless sensor networks because the additional overhead is proportional to the number of hops. To deal with the overhead, we propose an opportunistic multipath routing that forecasts the required number of paths, as well as bifurcation based on opportunistic routing according to the reliability requirement. In the proposed scheme, an intermediate node is able to select a different node for each transmission and to handle path failure adaptively. Through a performance evaluation, we demonstrate that the proposed scheme achieves a higher packet delivery ratio and reduces the energy consumption by at least approximately 33% and up to approximately 65% compared with existing routing protocols, under the condition of an 80% link success ratio in the long-hop sensor network.

## 1. Introduction

Most existing routing protocols in Wireless Sensor Networks (WSNs) are designed based on the single-path routing protocol. The single-path routing protocols could discovery the route with minimal computational complexity, resource utilization, and achievable network throughput. However, due to the unreliability of wireless links, single-path routing protocols cannot be considered effective techniques to meet the requirement like packet delivery ratio. Also, when the path fails to transmit a packet, extra overhead and delay occur to recover the path to deliver the packet continuously. Multipath routing protocols have the opposite advantages and disadvantages of single-path routing [1]. Both single-path and multipath protocols have their advantages and disadvantages; many studies have been proposed to satisfy the packet delivery ratio that is one of the important metric in WSNs.

Multipath routing protocols are exploited to various objectives such as load balancing [2,3] and quality of service (QoS) supporting [4,5]. These multipath focus on prolonging the network life time and qualified services. However, if a packet transmission failure occurs, it is not much different from a single-path because a packet is transmitted over only one path among the multiple paths. Thus, multipath routing protocols [6,7,8,9,10] could achieve the required packet delivery ratio by creating multiple paths to the destination and simultaneously transmitting the packet on the paths. In this paper, we focus on the last design issue to improve packet delivery ratio. In addition to multipath routing, the opportunistic routing protocols [11,12,13] are exploited to improve link reliability. As the opportunistic routing is a receiver-based transmission, an intermediate node not only selects one neighbor node to transmit data but also provides the transmission opportunity to all neighbor nodes. It could maximize a link reliability. The maximized link reliability could have a positive effect on the packet delivery ratio.

However, the packet delivery ratio would be degraded in a long-hop sensor network since packet or bit errors accumulate and the packet delivery ratio is the product of the link success ratio (assuming no retransmission) [14,15]. Thus, multipath routing requires larger numbers of paths to prevent the degradation of the packet delivery ratio of each path where the long-hop path is assumed in WSNs. That is, the existing multipath routing could satisfy the required packet delivery ratio by increasing the number of paths. However, the approach should accompany excessive energy consumption. In addition, sufficient numbers of paths are not feasible when the network is narrow. On the other hand, opportunistic routing cannot guarantee the required packet delivery ratio and requires more energy than single path routing because an intermediate node broadcasts the packet to all of its neighbors. Thus, opportunistic routing, in particular, suffers from packet delivery ratio degradation under long-hop WSNs.

Based on the deficiency of high-energy consumption in the abovementioned routing protocols, a sleep scheduling approach based on learning automata [16] has been proposed to solve the partial coverage problem, in which monitoring only a limited percentage of it is enough instead of monitoring the full coverage of WSNS. In other words, the proposed approach finds a minimal set of nodes to cover the demand portion based on learning automata with preservation of the connectivity among the nodes. As the number of participating nodes for monitoring is reduced, the energy consumption is reduced and the network lifetime is prolonged. However, since the multipath routing and opportunistic routing exploit a large number of nodes to improve packet delivery ratio, the decrease of participating nodes might degrade the performance of them. To improve reliability, reliable routing based on a learning algorithms [17,18] have been recently proposed. In these schemes, a source node selects an arbitrary node in every packet transmission to the destination according to the appraisals of each path, from the source node to the destination in the learning phase. Through the appraisal, the source node finds the best path with the highest packet delivery ratio to transmit the packet. Although this scheme could transmit the packet through the best path found in the learning phase, it does not prevent the degradation of the packet delivery ratio in a long-hop sensor network because it exploits only one path.

To address these problems, we propose an opportunistic multipath routing to improve the packet delivery ratio with low energy consumption in long-hop WSNs. In the proposed scheme, an intermediate node forecasts a required number of paths based on improved link reliability by the opportunistic routing. As an improvement to the link reliability leads to a packet delivery ratio, the proposed scheme could satisfy the required packet delivery ratio with a fewer number of paths compared with the existing multipath routing. To construct the multipath, the proposed scheme grants the transmission opportunity to the neighbor nodes of the intermediate node and the nodes with the required number of paths are selected. As the bifurcation process is performed locally, it could flexibly cope with the path failure problem of existing multipath routing protocols caused by exploiting the fixed paths to the destination. Through simulation, the proposed scheme improves the packet delivery ratio by at least 25% and up to 70%. In addition, the energy consumption is reduced by at least 33% and up to 65% compared with existing routing protocols under the 80% link success ratio in the long-hop sensor network.

The remainder of this paper is organized as follows. In Section 2, we describe the existing research on improvements of the packet delivery ratio. In Section 3, we explain the proposed multipath routing using opportunistic routing for energy efficiency and reliability in long-hop WSNs. The performance evaluation results are provided in Section 4. Finally, the proposed scheme is summarized and simulation results are presented.

## 2. Related Works

In this section, we briefly summarize the existing research relating to improvements of the packet delivery ratio: multipath, opportunistic, and learning-based routing protocols.

Multipath multi-SPEED protocol (MMSPEED) [6] is proposed to guarantee packet delivery services for reliability and real-time WSNs. To provide real-time services, the nodes select a network node with the proper speed of the packets and exploit the probabilistic multipath forwarding according to the requirements. It also performs without the global network information and could preserve various advantages, such as scalability for long-hop networks and flexibility with respect to network dynamics. However, to exploit this protocol in long-hop networks, it consumes massive amounts of energy to branch many paths to satisfy the packet delivery ratio. The robust and energy efficient multipath routing protocol (REER) [7] constructs the multipath paths in three phases: initial phase, main path construction, and sub-path construction. However, as  REER only constructs two paths, without considering the size of the network or the success rate of the nodes, REER could potentially suffer from packet delivery ratio degradation in long-hop networks. Lee et al. proposed a radio-disjoint geographic multipath routing protocol (RGMR) [8] that can determine the number of paths by the requirements. Thus, RGMR could satisfy the packet delivery ratio by transmitting the packet by multipath. However, it requires a high energy consumption because many paths are required to satisfy the required packet delivery ratio. In addition, as it should occupy a wide area to construct a radio-disjoint multipath, it may not be possible to construct a sufficient number of paths in a narrow network. Reliable and energy-efficient data collection protocol (RMER) [9] was proposed to meet the reliability and energy efficient requirements. In particular, it has the advantage of improving the network lifetime. To satisfy the reliability requirement and to reduce energy consumption, RMER evenly exploits the energy of the network nodes during packet transmission. In addition, the packets are aggregated near the hotspot area and transmitted to the sink. Therefore, RMER can maximize the network lifetime through the multipath. However, each packet is delivered through only one path and it may fail to transmit the packet close to the hotspot area when the scale of the network is increasing. Even the aggregated packet could fail to transmit to the sink. Energy-efficient reliable multipath data transmission (ERMDT) [10] is based on congestion avoidance. When congestion occurs during packet transmission on the main path, the packet is transmitted to the sink by an alternative path. In addition, it grades the priority of a packet. Thus, a packet with a high priority is transmitted first.

In opportunistic routing [11], which is a receiver-based routing protocol, a source node broadcasts the packet to all of its neighbors. The neighbor node, which receives a packet from the source node, has a transmission opportunity with a timer. The duration of the timer is decided by certain criteria such as the distance between itself and the destination. When the timer expires, the node broadcasts the packet to all of its neighbors, similar to the source. The neighbor node has a transmission chance with a timer, and the timer of the competitors are canceled. Adaptive opportunistic routing (AOR) protocol [12] was proposed for reliable data transmission. AOR also grants transmission opportunities to all of the neighbors with the energy and distance of the criterion. The node is at a shorter distance, has a higher energy, and can acquire the highest priority. To ensure that all neighbors recognize the packet transmission from the high-priority node to avoid duplicate transmission, AOR computes the packet holding time. However, AOR does not significantly differ from existing OR. As it could only increase the link reliability, AOR may suffer from degradation of the packet delivery ratio under long-hop networks. Reliability and real-time requirements based on opportunistic routing (RTGOR) [13] have also been proposed. RTGOR exploits the expected delivery reliability and transmission delay as the criterion. Similar to AOR, the node has a higher expected delivery reliability compared with that of a certain criterion and a lower transmission delay of the packet could acquire the highest priority. Through the reliability metric, the intermediate node filters out neighbor nodes that cannot satisfy the criteria. The node then exploits the delay as the other criteria to sort the remaining neighbor nodes. RTGOR could satisfy the reliability and real-time requirements in a small-sized network. However, the packet delivery ratio of RTGOR also decreases when the network scale is increased.

In the learning algorithm [19], a node constructs the best routing path through machine learning. In the fault tolerant reliable protocol (FTRP) [17], the nodes are in learning mode and broadcast a status of not being in a sensor domain in preparation to join one. If the node has not yet joined a domain and if it is not a member of the cluster, the node receives an answer from the sink node to be a cluster head. Otherwise, if the node receives an answer from some cluster head, the node will join the cluster. After constructing a cluster, the member node delegates the packet to its designated cluster head. The cluster head stores the packet until it receives an acknowledgement (ACK) message from a sink node. When it receives an ACK message for its previously sent packet, the packet is considered lost and retransmitted to ensure reliability. However, since it ensures reliability between the cluster head and the sink node by retransmission, it is hard to ensure the packet delivery ratio when the network scale is increased. This phenomenon would cause a buffer overflow on the cluster head because of the manystored packets retransmitted. The Energy-efficient algorithm for reliable routing (RRDLA) [18] considers the dynamics of links in finding a path from a source node to a destination by the QoS constraints such as end-to-end reliability. In the learning process, all of the paths from source to destination are evaluated. The source node finds the best path after finishing the process. The source node learns the best path and transmits the packet through the best path in the transmission process. The RRDLA could find the best path from the source node to the destination under QoS constraints through the learning algorithm. However, as the RRDLA exploits only one path to transmit the packet, it suffers from a degradation of the packet delivery ratio, even though the RRDLA selects the best path. That is, for a wireless sensor network in a multi-hop fashion, augmentation of end-to-end distance leads to an increased hop count. As a result, the augmentation of distance causes a degradation of the packet delivery ratio.

Table 1 summarizes the key features with a comparison of the reliability, energy consumption, and scalability of the related studies. MMSPEED and RGMR could satisfy the packet delivery ratio by increasing the number of paths according to the requirement. However, other protocols have difficulties in achieving the required packet delivery ratio because they exploit a static number of paths. As the energy consumption is heavily influenced by the number of paths, MMSPEED and RGMR consume more energy than the other protocols. Finally, with respect to the scalability, most studies cannot achieve the required packet delivery ratio because the overhead is proportional to the number of hops. In Section 3, a proposed scheme is described that satisfies the packet delivery ratio with a lower energy under long-hop sensor networks.

## 3. Energy-Efficient Reliable Multipath Routing Protocol Using Opportunistic Routing

In this section, we describe energy-efficient and reliable multipath routing using opportunistic routing to satisfy the required packet delivery ratio in long-hop WSNs. The proposed scheme can be largely categorized as a bifurcation decision, bifurcation transmission, and exceptional handling process.

### 3.1. Overview

In this section, we present the basic operation mechanism of the proposed routing protocol. The nodes are deployed over a field, and they could know a time *T* that has elapsed since the operation of a node began. Each node is aware of its own location by GPS or other techniques and knows the location of the destination. Moreover, each node keeps information about the location of its one-hop neighbor nodes by beaconing. The beacon interval $T_{B}$ is predetermined before network deployment according to the requirement of the applications. As shown in Figure 1, a packet from a source node is transferred to the destination performing path branching and merging on the intermediate nodes. The source and each intermediate node predict the packet delivery ratio from itself to the destination with opportunistic routing, firstly. If the predicted packet delivery ratio is lower than the required packet delivery ratio, the node calculates the number of paths and performs path bifurcation to satisfy the required packet delivery ratio (see more details in Section 3.2). In Figure 1, the nodes with diagonals perform the path bifurcation. If the node could satisfy the required packet delivery ratio without path bifurcation, the node transmits the packet based on opportunistic routing only (a white node on the transport path in Figure 1). As the nodes receive packets through opportunistic routing, they should compete to determine the transmitting node. In this process, each node starts a timer according to the distance between itself to destination. The shorter the distance, the shorter the duration of the timer. The node where the timer is expired transmits a packet, and its competitors quit the packet transmission process if a sufficient number of paths have been constructed (see more details in Section 3.3). Unfortunately, as transfers are by opportunistic routing, the path merger problem could occur (a black node on the transport path in Figure 1). In this case, assigned packet delivery ratios on each path would be merged and it leads to degradation of the overall packet delivery ratio required by the source node. To address this issue, the node aggregates the assigned packet delivery ratio from each parent node and exploits the aggregated packet delivery ratio when it transmit the packet to its destination (see more details in Section 3.4).

### 3.2. Multiple Path Bifurcation Decision Process

In this section, we describe the path bifurcation decision process to determine the number of paths. Algorithm 1 describes the pseudocode for the bifurcation decision process at an intermediate node. The process proceeds as follows: (1) predict the packet delivery ratio, (2) decide whether to perform the bifurcation, and (3) calculate the number of paths. $P_{n}$ is the link success ratio to node *n*, and *k* is the number of neighbors that are located closer to the destination compared with the intermediate node.

**Algorithm 1** Pseudocode for the bifurcation decision process at an intermediate node *i*
1:**Input**: $P_{req}$ ← required packet delivery ratio, $k_{i}$ ← the number of neighbors2:**Output**: *n* ← the number of paths(default = 1)3: 4:$P_{\left(k_{i},1\right)}=1-\left(1-P_{1}\right)\left(1-P_{2}\right)\ldots \left(1-P_{k_{i}}\right)$       ▹ link reliability by opportunistic routing5:$N_{hop}$ = $⎡\frac{Dist(i,destination)}{r}⎤$       ▹ expected hop-count from itself to the destination6:$P_{exp}$ = ${\left(P_{\left(k,1\right)}\right)}^{N_{hop}}$                   ▹ expected packet delivery ratio7: 8:**if**$P_{exp}>P_{req}$**then**               ▹ bifurcation do not require9:    return *n*10:
**end if**
11: 12:**while**$P_{exp}<P_{req}$**do**               ▹ bifurcation required13:    *n* = *n*+114:    $P_{exp}$ = ${\left(P_{\left(k_{i},n\right)}\right)}^{N_{hop}}$     ▹$P_{exp}$ when *n* of $k_{i}$ nodes is selected by opportunistic routing15:
**end while**
16: 17:return *n*


**Step 1 (lines 4–6):** First, all nodes can measure the link quality of each other through beacon message exchange before data transmission. Each node knows the transmission cycle of beacon messages; the link quality is calculated through the number of received beacon messages from each neighbor node over some time. For example, if the transmission cycle of beacon message is 1 second and an intermediate *i* receives 80 beacon messages for 100 seconds from a neighbor node *j*, the link quality $P_{j}$ is 80%. That is, the link quality $P_{j}$ could be computed as follows:(1)$$P_{j}=\frac{B_{recv}}{\left(T/T_{B}\right)}$$
where (Equation 1), $B_{recv}$ is a number of received beacon message and *T* and $T_{B}$ are simulation time and beacon duration, respectively.

Based on the measurements, an intermediate node can acquire the link quality to each neighbor node and forecast the link reliability $P_{\left(k_{i},1\right)}$ based on opportunistic routing as $P_{\left(k_{i},1\right)}=1-\left(1-P_{1}\right)\left(1-P_{2}\right)\;\ldots \;\left(1-P_{k_{i}}\right)$ (line 4). Let *r* be the radius of the radio range and Dist(a,b) be the distance from *a* to *b*. The intermediate node calculates expected number of hop counts from itself to the destination $N_{hop}$ as ⎡Dist(itself,destination)/r⎤ (line 5). Using the probability $P_{\left(k_{i},1\right)}$ and the expected number of hop counts $N_{hop}$, the intermediate node can obtain the expected packet delivery ratio ($P_{exp}$) given by ${\left(P_{\left(k_{i},1\right)}\right)}^{N_{hop}}$ (line 6).

**Step 2 (lines 8–10):** After predicting the packet delivery ratio, the intermediate node compares $P_{exp}$ with the required packet delivery ratio ($P_{req}$). If the $P_{exp}$ is greater than $P_{req}$, the intermediate node decides whether path bifurcation is required and returns *n* (lines 8–10).

**Step 3 (lines 12–17):** However, if $P_{exp}$ is less than $P_{req}$, the intermediate node decides that path bifurcation should be performed to improve $P_{exp}$ (line 12). The intermediate node increases *n* by 1 and recalculates $P_{exp}$${\left(P_{\left(k,n\right)}\right)}^{N_{hop}}$ (lines 13–14). When the intermediate node determines the number of nodes required to satisfy the condition that $P_{exp}$ is greater than $P_{req}$, the intermediate node can find the number of paths, *n*, to satisfy $P_{req}$ and returns *n* (line 17).

However, during this process the issue of whether a neighbor node not realizing a packet transmission of the other neighbor nodes may arise. That is, if the distance between a pair of neighbor nodes of the intermediate node is longer than the transmission radius of the sensor, they may not recognize the transmission of the other, causing excessive bifurcation without timer termination. Figure 2 shows the node selection problem. When the intermediate node, *S*, broadcasts the packet to its neighbor nodes, the neighbor nodes start the timer to broadcast the packet. Suppose that a timer of node 1 expires when the number of calculated paths is one. Node 1 tries to broadcast the packet, and the other neighbor nodes terminate their timers when they receive the packet of node 1. However, nodes 4 and 5 are located outside the transmission range of node 1. Thus, nodes 4 and 5 cannot overhear the packet of node 1 and the timer does not terminate. Finally, nodes 4 or 5 broadcast the packet when their timers have expired. This phenomenon causes excessive bifurcation.

To solve this problem, the intermediate node calculates a neighbor set that selects the maximum number of neighbors. The nodes in the set should be located within a communication radius of each other. That is, the node can communicate with any node in the set. To calculate the neighbor set, the intermediate node calculates the center point of all of the neighbor nodes and excludes the farthest node from the center point. The intermediate node then measures the distance between the farthest pair among the neighbor nodes. If the measured distance is less than the transmission radius, the neighbor set calculation is completed. However, if the distance is greater than the radius, the abovementioned process is repeated until the proper neighbor set is found. Through this process, the intermediate node can find the proper neighbor set in which all of the nodes are separated by the communication radius. As shown in Figure 2, the source *S* calculates the center point *C* of all of the neighbor nodes. The source node excludes node 1 that is the farthest from *C* among the neighbor set. The source node measures whether the two most distant neighbors of the remaining neighbors, nodes 2 and 5, are within the communication radius of each other. As the distance is shorter than the transmission radius *r*, nodes 2–5 are located within one communication radius of each other. The source node *S* forms a neighbor set consisting of nodes 2–5. Finally, as the node *S* broadcasts the packet to the formed neighbor set, opportunistic routing could avoid the abovementioned problem.

### 3.3. Bifurcation Transmission Process Based on Opportunistic Routing

After calculating the number of paths through the path bifurcation decision process, the intermediate node transmits the packet to the destination. In this section, we explain the bifurcation transmission process for a reliable packet transmission. Existing opportunistic routing provides a transmission opportunity to only one neighbor node. However, the proposed scheme provides a transmission opportunity to the same number of neighbor nodes as the number of paths determined in the path branch decision process. To grant an opportunity to a number of neighbors, the intermediate node records the number of paths in the packet. The neighbor nodes can know how many neighbors will be selected. Algorithm 2 describes the bifurcation transmission process to transmit the packet as many as the number of paths from the point of view the receiver. The process consists of two cases: (1) expiring timer and (2) receiving a number of transmissions from other competitors. If the intermediate node decides to bifurcate to *n* paths, the intermediate node records the bifurcation number *n* in the packet and broadcasts it to the neighbor nodes (line 1). The timer of the neighbor nodes that received the packet would expire sequentially depending on the distance from itself, *j*, to the destination of the opportunistic routing (line 7). The waiting time *t* is calculated as follow. Each neighbor node *j* receives a message from intermediate node *i* and could predict the distance to the  destination and the expected number of hop counts. The node *j* divides the distance by the expected number of hop counts and exploits the result as a criterion for waiting time. For example, if the distance from arbitrary node *j* to the destination is 500 m and the expected number of hop counts is 5, the waiting time for node *j* could be 100 ms. That is, the waiting time $T_{j}$ of node *j* could be computed as follows:(2)$$T_{j}=\frac{D_{\left(j,Dest\right)}}{H_{\left(j,Dest\right)}}\times 10^{k}ms$$
where $D_{\left(j,Dest\right)}$ is a distance between *j* to the destination and $H_{\left(j,Dest\right)}$ is the number of expected hop counts from *j* to the destination as $⎡D_{\left(j,Dest\right)})/r⎤$. *k* is exploited to modulate the time duration according to node specifications, network congestion, and various points of consideration for requirements. A smaller *k* value leads to a shorter delay to a transmit packet, while a larger *k* value could mitigate network congestion.

**Case 1 (lines 9–14):** Suppose a timer of an arbitrary node, *j*, is expired (line 9); then, the node calculates the number of paths, $n^{\prime }$, by Algorithm 1 (line 10) and inserts $n^{\prime }$ into the packet $p_{j}$ (line 11). The intermediate node, *j*, then broadcasts the packet $p_{j}$ to its neighbor nodes (line 12) and quits the competition process (line 13).

**Case 2 (lines 15–22):** Unlike case 1, node *j* could receive packets from other competitors before the timer expires (line 15). In this case, node *j* counts the number of transmissions overhearing *h* (line 16). If *h* is greater than *n* (line 17), then the node *j* realizes that a sufficient number of paths have been constructed through other competitors. Thus, node *j* terminates the timer *T* (line 18) and drops the packet $P_{j}$ (line 19) to prevent unnecessary bifurcation.

**Algorithm 2** Pseudocode for the receiver-based transport competition process at neighbor node *j*
1:**Input**: $p_{i}$ ← received packet from the intermediate node *i*2: 3:$p_{j}$ ← the packet to transmit at node *j*4:*h* ← the number of overhearing (default = 0)5: 6:*n* = GetNumberofPaths($p_{i}$)        ▹ extract the number of paths *n* from the packet $p_{i}$7:StartTimer(*T*)        ▹ timer for waiting according to distance from *j* to destination8:**while** TRUE **do**9:    **if**
*T*.expired() == TRUE **then**            ▹ waiting timer expiration10:        $n^{\prime }$ = bifurcation_decision_process$(P_{req}$, $k_{j})$ ▹ calculate the number of path $n^{\prime }$ (Algorithm 1)11:        $p_{j}$.SetNumberofPaths($n^{\prime }$)        ▹ set the number of paths $n^{\prime }$ to satisfy the $R_{req}$12:        Send($p_{j}$)                        ▹ send a packet $p_{j}$13:        Exit()                      ▹ quit the competition process14:    **end if**15:    **if** Receive($p_{j}$) == TRUE **then**      ▹ when receive the packet $p_{j}$ from other competitor16:        *h* = *h* + 117:        **if**
*h* ≥ *n*
**then**           ▹ already enough number of paths are constructed18:           *T*.Stop()                ▹ stop timer to transmit the packet19:           Drop($p_{j}$)                    ▹ drop the packet $p_{j}$20:           Exit()                  ▹ quit the competition process21:        **end if**22:    **end if**23:
**end while**



In addition, we should consider the distribution of $P_{req}$ to prevent unnecessary bifurcation. Suppose that each bifurcation path has the same value as $P_{req}$ of the intermediate node; then, the total $P_{req}$ increases, leading to an unnecessary bifurcation. For example, consider an application requiring a $P_{req}$ of 80% with a bifurcation number 2. If the intermediate node does not distribute $P_{req}$ to each path, each path attempts to achieve a $P_{req}$ of 80%. That is, the total $P_{req}$ is 96%, i.e., 1−(1−0.8)(1−0.8). Each node on the path can also perform an unnecessary bifurcation to satisfy $P_{req}$ when $P_{req}$ is not properly distributed.

Figure 3 shows an example of the bifurcation transmission process and $P_{req}$ distribution to prevent unnecessary bifurcation. To explain the bifurcation process and $P_{req}$ distribution, we suppose that the source node, *S*, and node 3 are decided to perform the branching with two paths and the original $P_{req}$ at the source node is 80%. To bifurcate the path, the source node *S* calculates the proper distribution of $P_{req}$. In Figure 3, the source node distributes a $P_{req}$ of 80% to 50% and 60% $P_{req}$. The source node *S* records the number of paths and distributed $P_{req}$ in the packet and broadcasts the packet. Nodes 1 and 3 receive the packet from the source node *S*. A timer of node 3 that is closer to the destination expires before the timer expiration of the node 1. At that time, node 3 realizes the given $P_{req}$ is 50%. In the case of node 1, as for the timer of node 3, node 1 knows that only one node has a transmission opportunity with 60% $P_{req}$. When a timer of node 1 expires, node 1 broadcasts the packet with the 60% $P_{req}$. When $P_{req}$ that is distributed to nodes 1 and 3 are combined, the calculated total $P_{req}$ is 80%, i.e., 1−(1−0.5)(1−0.6). In other words, even if the path is bifurcated, the original $P_{req}$ remains unaffected and the proposed scheme can prevent unnecessary branching. If node 3 requires bifurcation of the path, since $P_{exp}$ of the node 3 is less than 50% $P_{req}$, it is possible to construct an additional multipath by further distributing $P_{req}$ to nodes 5 and 6, as shown in Figure 3.

### 3.4. Exception Handling

As opportunistic routing is only aimed at the probabilistic increase of the link reliability by giving an opportunity to all of the neighbor nodes, all nodes may fail to transmit the packet or else a sufficient number of paths cannot be constructed. In this case, the packet delivery ratio is degraded. To complement this problem, we exploit a characteristic of the opportunistic routing called overhearing. We present an exception handling process (Algorithm 3). When the intermediate node broadcasts the packet to its neighbor nodes, the intermediate node starts a timer to overhear the transmission of the neighbor nodes and counts the number of transmissions (lines 6–10). Similar to the competition process (Algorithm 2), the exception handling process is divided into three cases: (1) all of the paths are successfully constructed, (2) no paths are constructed, or (3) only some paths are constructed.

**Case 1 (lines 11–13):** If the intermediate node *i* received the same number of transmissions *h* as the number of paths *n* (Line line 11), the intermediate node realizes that a sufficient number of paths are constructed that satisfy $R_{req}$. In this case, the exception handling process is terminated without any handling because a sufficient number of paths have already been created (line 12).

**Case 2 (lines 17–19):** On the other hand, the intermediate node *i* cannot receive any packet $p_{j}$ (line 17). This phenomenon means that all of the neighbor nodes fail to receive the packet $p_{i}$ from the intermediate node *i*. Thus, the intermediate node *i* retransmits the packet $p_{i}$ to the neighbor nodes to construct paths (line 18).

**Case 3 (Line lines 20–23):** Similarly, in the case in which the intermediate node *i* overhears a lesser number *h* of transmissions than the number of calculated paths *n* (line 20), the intermediate node *i* judges that an insufficient number of paths has been constructed. The intermediate node *i* calculates the required number of paths n−h to construct *n* paths and inserts n−h into the packet $p^{\prime }$ (line 21). The intermediate node *i* then transmits the packet $p^{\prime }$ to the neighbor nodes (line 22).

The modified packet includes the number of modified paths h−n and modified $P_{req}$. For example, suppose that three paths are required to satisfy $P_{exp}$ and the intermediate node overhears only one transmission. In this case, the intermediate node modifies the packet, the number of paths is reduced from 3 to 2, and the distributed $P_{req}$ is also reduced. Through this process, the proposed scheme can prevent the degradation of the packet delivery ratio by a failure of the transmission based on the opportunistic routing.

**Algorithm 3** Pseudocode for an exceptional case handling process at an intermediate node *i*
1:*n* ← number of paths        ▹ number of paths calculated by Algorithm 12:$p_{i}$ ← original packet to construct the number of paths at node *i*3:$p^{\prime }$ ← modified packet to handle the exception4:$p_{j}$ ← received packet from the neighbor node *j*5:*h* ← number of overhearings (default = 0)6: 7:StartTimer(*T*) ▹ when this timer expires, the timer of all neighbors is considered to have expired8:**while***T*.expired() == FALSE **do**     ▹ overhearing the transmission of the neighbors9:    **if** Receive($p_{j}$) == TRUE **then**   ▹ when receiving the packet $p_{j}$ from the neighbor nodes10:        *h* = *h* + 111:        **if**
h=n
**then**                  ▹ a sufficient paths are constructed12:           Exit()                  ▹ quit the exception handling process13:        **end if**14:    **end if**15:
**end while**
16: 17:**if***h* = 0 **then**                         ▹ no paths were constructed18:    Send($p_{i}$)19:
**end if**
20:**if***h* < *n*
**then**  ▹ Some paths were constructed and the number of paths is insufficient by n−h.21:    $p^{\prime }$.SetNumberofPaths(n−h)    ▹ set the number of paths, n−h, to construct *n* paths22:    Send ($p^{\prime }$)          ▹ modified packet is retransmitted23:
**end if**



In addition to the transmission failure problem, the path merger problem occurs because a proposed scheme cannot guarantee a node-disjoint transmission. That is, an arbitrary node can receive two or more packets from different parent nodes. To deal with this problem, when the node receives two or more packets, the node finds the $P_{req}$ of each packet. Then, the node combines the $P_{req}$ of each packet into one $P_{req}$. The node calculates the required number of paths with a combined $P_{req}$ when it transmits the packet to the destination. Figure 4 shows an example of the path merger problem. Suppose that the intermediate node 1 performs a branch to nodes 2 and 3 with 30% of $P_{req}$. Suppose also that node 4 is selected as the opportunistic routing results of nodes 2 and 3. In this case, node 4 finds the $P_{req}$ of each packet form from nodes 2 and 3. In the case of Figure 4, the $P_{req}$ of nodes 2 and 3 is 30%. Thus, node 4 merges $P_{req}$, i.e., 1−(1−0.3)(1−0.3). As a result, node 4 can obtain approximately 50% of $P_{req}$. A comparison of $P_{req}$ of the intermediate node 1 and calculated $P_{req}$ of node 4 confirms that node 4 can obtain the same value through the merger process. That is, even if the path merging problem occurs in the bifurcation process, the proposed scheme can prevent the degradation of $P_{req}$.

## 4. Performance Evaluation

In this section, we describe the simulation results for the REER, RGMR, RRDLA, and the proposed scheme. REER is the most representative multipath routing protocol that constructs two paths and transmits data on the paths. RGMR constructs several paths depending on the required packet delivery ratio. RRDLA selects the best single path based on the learning algorithm.

In Section 4.1, we analyse a time complexity to confirm that a node has a capability to perform the proposed scheme. In Section 4.2, we describe the simulation environment and evaluation factors. The packet delivery ratio versus the end-to-end distance and link reliability is described in Section 4.3. The energy consumption versus the end-to-end distance and link reliability are explained in Section 4.4. The packet delivery ratio and energy consumption versus the number of source nodes are explained in Section 4.5. Finally, The packet delivery ratio and energy consumption versus the number of nodes are described in Section 4.6.

### 4.1. Time Complexity Analysis

In this section, to verify the capability of the node to perform the proposed scheme, we present the time complexity analysis results. Table 2 summarizes the time complexity of the proposed scheme and other protocols. The proposed scheme is formed by two nested loops: (1) the inner loop that has a running time proportional to the number of nodes (*N*) to calculate the required number of paths and (2) the outer loop that has a running time depending upon the number of paths (*P*). Therefore, the complexities of the inner and of the outer loops are equal to O(N) and O(P), respectively. In addition, the running time of the exceptional handling process is O(1) because the process would be terminated within a fixed time. As a result, the time complexity of the proposed scheme could be expressed as O(N×P)+O(1). Therefore, the overall time complexity of the proposed scheme is O(N×P). As the number of nodes *n* is much more than the number of paths *P*, the time complexity of the proposed scheme could be considered as O(N). Compared to the other protocols, REER has a running time proportional to the number of nodes (*N*) to calculate the link cost. Thus, the time complexity of REER is O(N). RGMR is formed by two nested loops: (1) the inner loop that has a running time proportional to the number of nodes (*N*) to calculate required number of paths and (2) the outer loop that has a running time according to number of paths (*P*) to calculate enter/exit point of each path. Thus, the time complexity of RGMR is O(N×P). Finally, the RRDLA also has two nested loops: (1) the inner loop that has a running time proportional to the number of nodes (*N*) and (2) the outer loop that has a running time to meet the condition (*K*). Thus, the time complexity of RRDLA is O(N×K).

### 4.2. Simulation Environments

We simulate and analyze the proposed scheme and other protocols on NS-3 simulator. Table 3 describes the detailed setup of our simulation. The nodes are placed in a 1000 × 1000 m terrain. As shown in Figure 5, 100 nodes are placed in the form of a grid and the remaining 900 are randomly placed. Thus, the node density is determined arbitrarily, which can lead to the exceptions described in Section 3.4. Especially, as the black square in Figure 5 has a low node density, it adversely affects the performance. The provided results have considered these simulation environments. According to the MICA2 specification, the transmission and receiving power consumption of the sensor node are 24.92 and 19.72 mJ per one byte. Using the simulation, we measure the packet delivery ratio and energy consumption in terms of the end-to-end distance and link reliability. The simulations are performed 30 times, and the provided graph represents the average result of the simulations. The evaluation factors and terms are summarized as follows:
–End-to-end distance is the shortest distance from the source node to the destination.–Link reliability is the average link transmission success ratio of the node.–The number of source nodes is defined as the number of source nodes that transmits a packet to the destination.–The number of nodes is defined as the number of nodes batched on the network.–Packet delivery ratio is the ratio of the number of data arriving at the destination to the number of generated by the source node.–Energy consumption is the average energy consumption of the nodes that attended the transmission process.

### 4.3. Packet Delivery Ratio Versus End-to-End Distance and Link Reliability

In this section, to investigate the effect of the link reliability on the packet delivery ratio, we present the performance evaluation results while increasing the end-to-end distance when the link reliability is 80% and 90%. In this simulation, the failed packets during the transmission process do not retransmit to confirm the change in the packet delivery ratio.

Figure 6 shows the packet delivery ratio versus the end-to-end distance when the link reliability is 90%. The proposed scheme could can satisfy the required packet delivery ratio by path bifurcation and high packet delivery ratio based on opportunistic routing. RGMR also satisfies the required packet delivery ratio based on several radio-joint paths generated by the required packet delivery ratio. However, REER and RRDLA cannot satisfy the required packet delivery ratio because they have a limitation on the number of paths. That is, two paths of REER are not sufficient to satisfy the required packet delivery ratio in a long-hop network. Similarly, although RRDLA could find the best path through the learning algorithm, it cannot satisfy the required packet delivery ratio in the long-hop network as it exploits only one path to transmit data.

The packet delivery ratio is further degraded when the link reliability is degraded. Figure 7 shows the packet delivery ratio versus the end-to-end distance when the link reliability is 80%. The packet delivery ratio is slightly degraded; however, the proposed scheme can satisfy the required packet delivery ratio for the abovementioned reasons. Similarly, RGMR can satisfy the required packet delivery ratio; however, the packet delivery ratio decreases when the scale of the network increases to greater than 800 m${}^{2}$. This is because RGMR cannot construct more paths to satisfy the requirement. That is, the radio-joint multipath scheme RGMR requires a wide path to prevent collisions and interference between each path. Owing to this, the packet delivery ratio is degraded because it cannot construct a sufficient number of paths. For the REER and RRDLA cases, the degradation of the packet delivery ratio is more severe. When the network size is 1000 m${}^{2}$, the packet delivery ratio of REER sharply drops from approximately 60% to approximately 20% for a link reliability of 90%. Likewise, the packet delivery ratio of RRDLA drops from approximately 34% to approximately 10% for a link reliability of 90%.

### 4.4. Energy Consumption Versus End-to-End Distance and Link Reliability

In this section, we demonstrate the effect of the link reliability and end-to-end distance on energy consumption. We perform an evaluation while increasing the end-to-end distance when the link reliability is 80% and 90%. To investigate the energy consumption when achieving the same packet delivery ratio, the failed packets during the transmission process are retransmitted.

Figure 8 shows the energy consumption versus the end-to-end distance when the link reliability is 90%. When the scale of the network size is small, RRDLA consumes the smallest amount of energy because it exploits only one of the best paths found by the learning algorithm. REER consumes the second least energy amount because it uses only two paths. The proposed scheme requires a higher energy than RRDLA and REER. This is because the proposed scheme grants transmission opportunities to all of the neighbor nodes. That is, as the number of receiving packets is greater than those of RRDLA and REER, the proposed scheme consumes more energy. RGMR consumes more energy than the other protocols because it constructs several paths to satisfy the required packet delivery ratio. In summary, with regard to energy consumption when the scale of the network size is small, the proposed scheme and RGMR could satisfy the required packet delivery ratio without packet retransmission. However, although RRDLA and REER perform packet retransmission to satisfy the required packet delivery ratio, the energy consumption of the retransmission is less than the energy consumption for multipath construction and transmission. However, as the scale of the network size increases, the proposed scheme consumes less energy than the other protocols. The reason for this is that there is little difference between the cases where bifurcation is performed or not based on opportunistic routing because the proposed scheme always grants all of the neighbors to the neighbor nodes. The energy consumption of RRDLA and REER increase depending on the increase in the number of retransmissions. RGMR consumes the most energy because it continues to increase the path to achieve the requirements.

Figure 9 shows the energy consumption versus the end-to-end distance when the link reliability is 80%. Similar to the case with a link reliability is of 90%, all of the protocols consume more energy as the end-to-end distance increases. However, the energy consumption of the proposed scheme is similar to that of a packet delivery ratio of 90%. This is because the proposed scheme scarcely retransmits the packet, courtesy of the characteristic of opportunistic routing, which improves the link reliability. However, for the other protocols, as the packet delivery ratio of each path is degraded, the ratio of the retransmission increases. In addition, RGMR cannot construct enough paths when the scale of the network size increases over 800 m${}^{2}$ and needs to perform retransmission. Thus, the energy consumption slowly increases compared with more path construction.

### 4.5. Packet Delivery Ratio and Energy Consumption Versus the Number of Source Nodes

In this section, we demonstrate the effect of the number of source nodes on the packet delivery ratio and energy consumption when the link reliability is 90%. Figure 10 shows the packet delivery ratio versus the number of source nodes. To investigate the variation in the packet delivery ratio, the failed packets do not retransmit. The packet delivery ratio of the protocols is degraded when the number of source nodes increases owing to the buffer overflow caused by frequently collision avoidance (especially nearby destination) by operation of carrier sense multiple access with collision avoidance (CSMA/CA). Thus, the packet delivery ratio of RGMR, which constructs the most paths, decreases the most. For the proposed scheme, the packet delivery ratio drops slightly due to congestion around the sink node. REER also suffers a slight decline in the packet delivery ratio because the number of paths increases. As RRDLA exploits only a single path to transmit the packet, less congestion and interference occur compared with the other protocols.

Figure 11 shows the energy consumption versus the number of source nodes. To investigate the energy consumption when achieving the same packet delivery ratio, the failed packets are retransmitted. When the number of source nodes increases, the energy consumption also increases. However, the increase is not proportional to the number of source nodes. In the case of RGMR, the protocol exploits the maximum number of paths to satisfy the required packet delivery ratio. However, as most of the constructed paths by different source nodes overlap, collisions and interferences occur. That is, as the maximum number of transmissions and retransmissions occur, RGMR consumes the most energy. As REER and RRDLA construct the lowest number of paths compared with the results of RGMR, they consume a much smaller amount of energy. However, they cannot avoid congestion around the sink node. This leads to an increase in the number of retransmissions and energy consumption. The proposed scheme also suffers from the same problem. Thus, energy consumption of the proposed scheme slightly increases due to the collisions and interference around the sink node.

### 4.6. Packet Delivery Ratio and Energy Consumption Versus the Number of Nodes

In this section, we demonstrate the effect of the number of nodes on the packet delivery ratio and energy consumption when the link reliability is 90%. Figure 12 shows the packet delivery ratio versus the number of nodes. To investigate the variation in the packet delivery ratio, the failed packets do not retransmit. The packet delivery ratio of the protocols is degraded when the number of nodes decreases because the decrease of node density disturbs multipath construction. Thus, the proposed scheme and RGMR, which construct a number of paths, are the most affected. On the other hands, as REER constructs only two paths, it maintains a similar packet delivery ratio with little effect by on the node’s density. In the case of RRDLA, when the node density is increased, the probability of obtaining a path consisting of nodes with high link reliability is increased. Thus, the packet delivery ratio of RRDLA slightly increases when the number of node is increased.

Figure 13 shows the energy consumption versus the number of nodes. To investigate the energy consumption when achieving the same packet delivery ratio, the failed packets are retransmitted. When the number of nodes decreases, the energy consumption is increased. Especially, the energy consumption of RGMR which constructs the number of paths is sharply increased. It is because RGMR requires a number of nodes to construct multipath; however, it cannot construct sufficient multipaths. Thus, RGMR continuously tries to construct multipaths and a large number of failed packets by path construction failure are retransmitted, increasing energy consumption rapidly. Similarly, the energy consumption of the proposed scheme is increased under the low-density network because it cannot secure enough number of nodes to exploit an advantage of the opportunistic routing. In the case of REER and RRDLA, there is a slight increase in energy consumption. The reason why they have not suffered from significant energy drain compared to RGMR and the proposed scheme is because when they perform packet retransmission to satisfy the required packet delivery ratio, the energy consumption of the retransmission is less than the energy consumption for path recovery.

## 5. Conclusions

Many studies have been proposed for the improvement of the packet delivery ratio as it is one of the most important factors in WSNs. However, the existing studies have not achieved a sufficient packet delivery ratio in a long-hop sensor network and some require excessive energy consumption to satisfy the requirement. Therefore, we propose an energy-efficient reliable routing protocol based on opportunistic routing. The proposed scheme maximizes the link reliability and bifurcates with only one broadcast based on the characteristic of opportunistic routing that grants a transmission opportunity to all of the neighbor nodes. In other words, the proposed scheme can reduce the number of required paths due to the maximized link reliability and can achieve a high packet delivery ratio with the least number of paths compared with the existing multipath schemes. Moreover, as each node performs a contention-based transmission, the proposed scheme can avoid the potential problems of multipath studies, such as path failure and wasted recovery cost, by exploiting the fixed paths. From the simulation results, the proposed scheme reduces the energy consumption from at least approximately 33% to up to approximately 65% and shows a higher packet delivery ratio compared with those of existing routing protocols under the condition with an 80% link success ratio in a long-hop network.

## Figures and Tables

**Figure 1 sensors-19-04072-f001:**
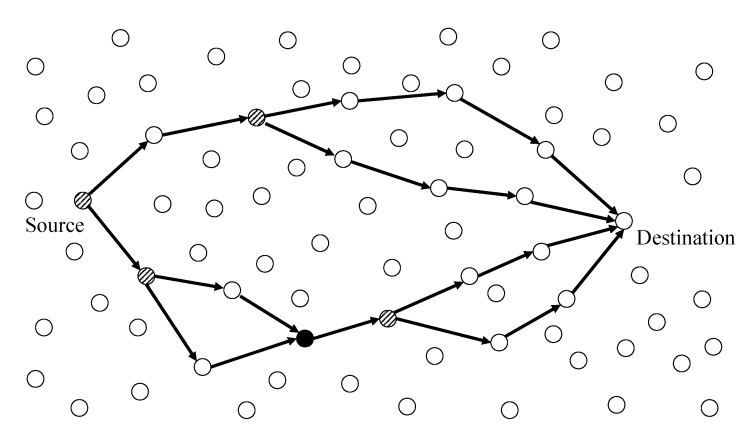
An overview of the opportunistic mulitpath routing protocol.

**Figure 2 sensors-19-04072-f002:**
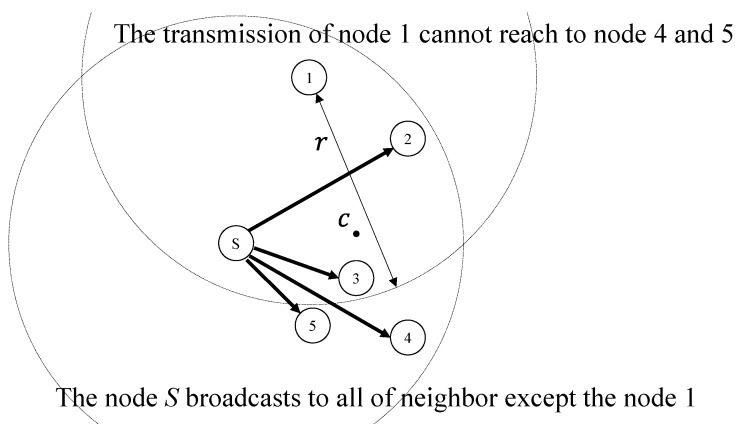
An example of the node selection problem.

**Figure 3 sensors-19-04072-f003:**
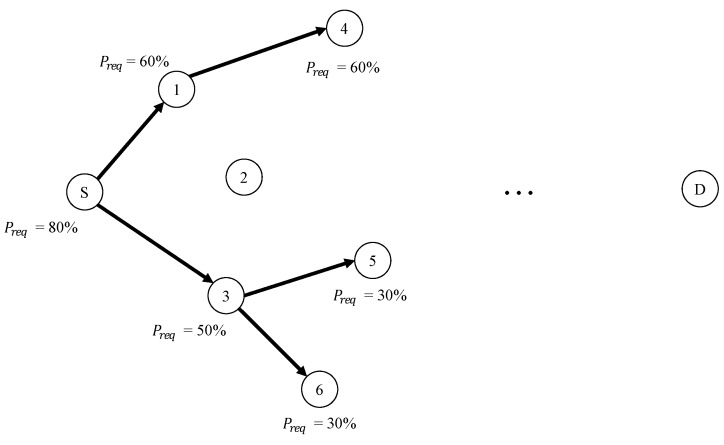
An example of the bifurcation transmission process and $P_{req}$ distribution.

**Figure 4 sensors-19-04072-f004:**
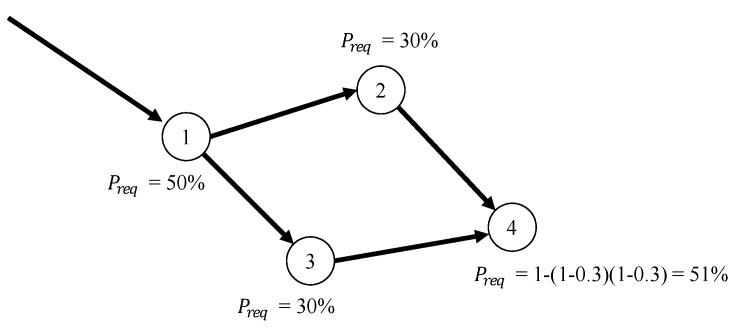
An example of the path merger problem.

**Figure 5 sensors-19-04072-f005:**
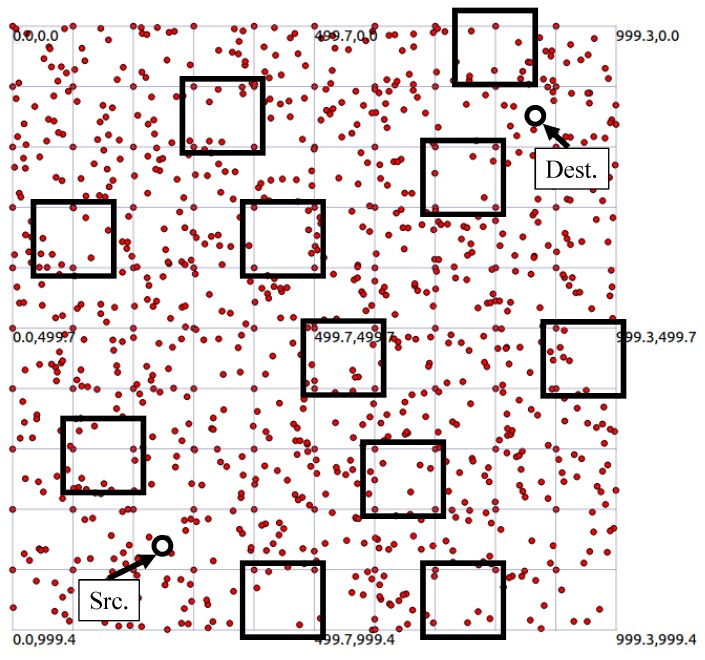
Simulation topology.

**Figure 6 sensors-19-04072-f006:**
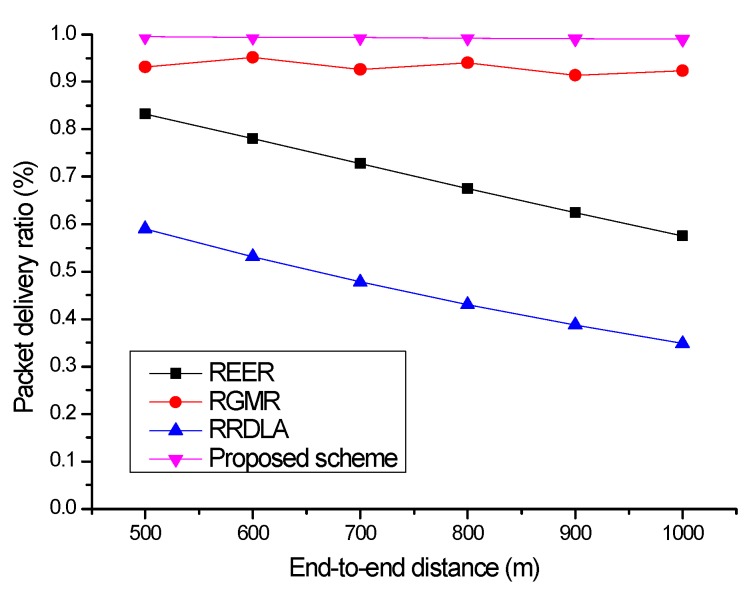
Packet delivery ratio vs. end-to-end distance with 90% of link reliability.

**Figure 7 sensors-19-04072-f007:**
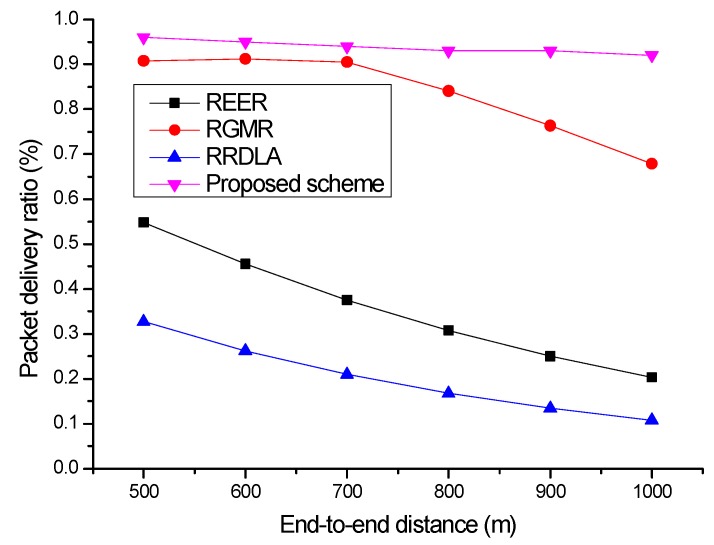
Packet delivery ratio vs. end-to-end distance with 80% of link reliability.

**Figure 8 sensors-19-04072-f008:**
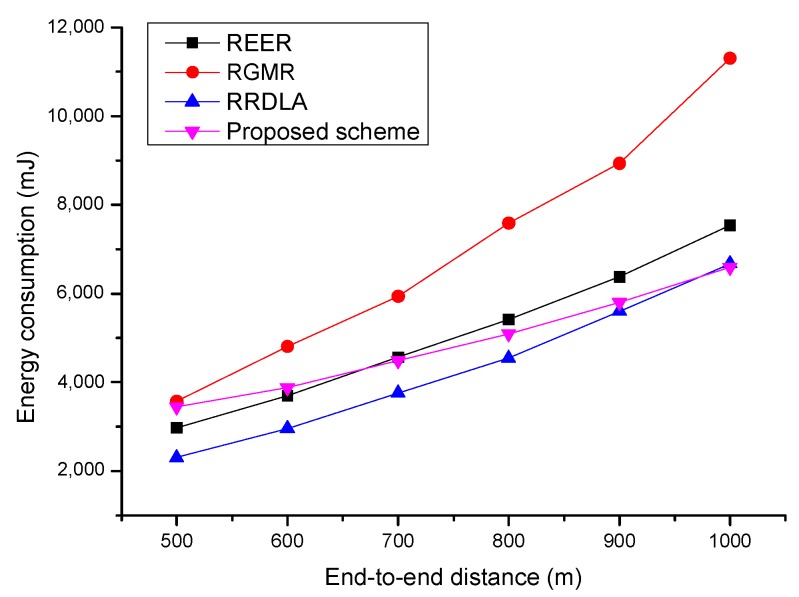
Average energy consumption vs. end-to-end distance with for 90% of the link reliability.

**Figure 9 sensors-19-04072-f009:**
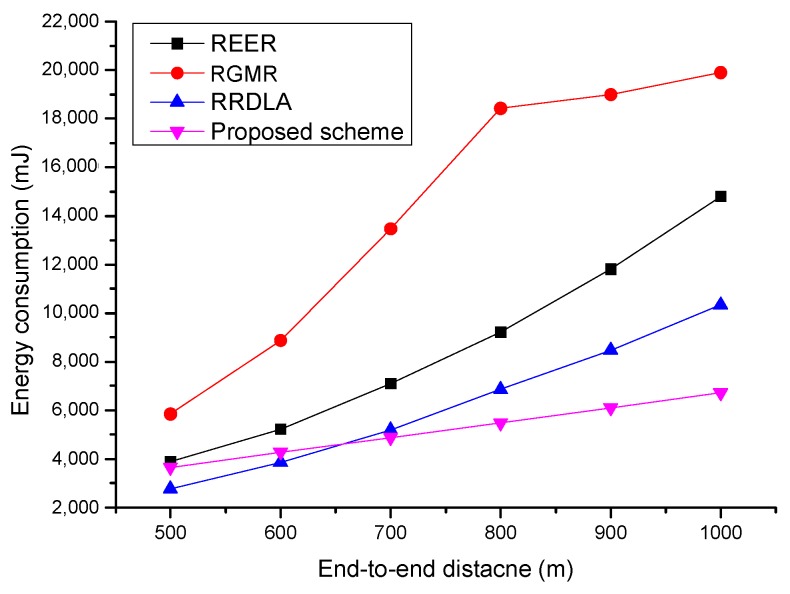
Average energy consumption vs. end-to-end distance with for 80% of the link reliability.

**Figure 10 sensors-19-04072-f010:**
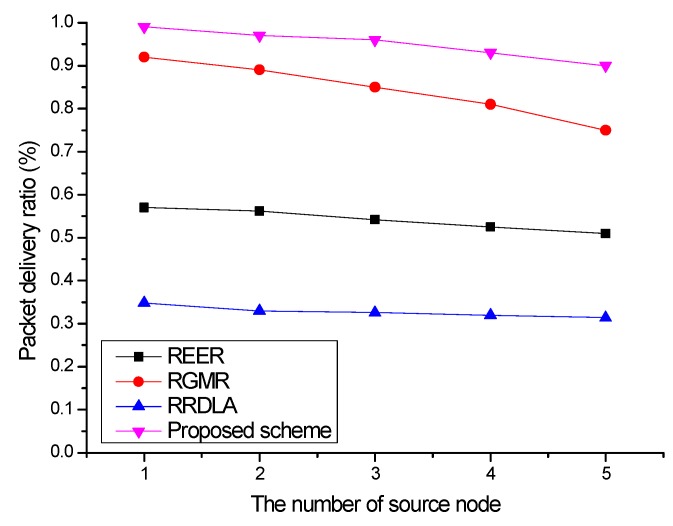
Packet delivery ratio vs. the number of source nodes with 90% of the link reliability.

**Figure 11 sensors-19-04072-f011:**
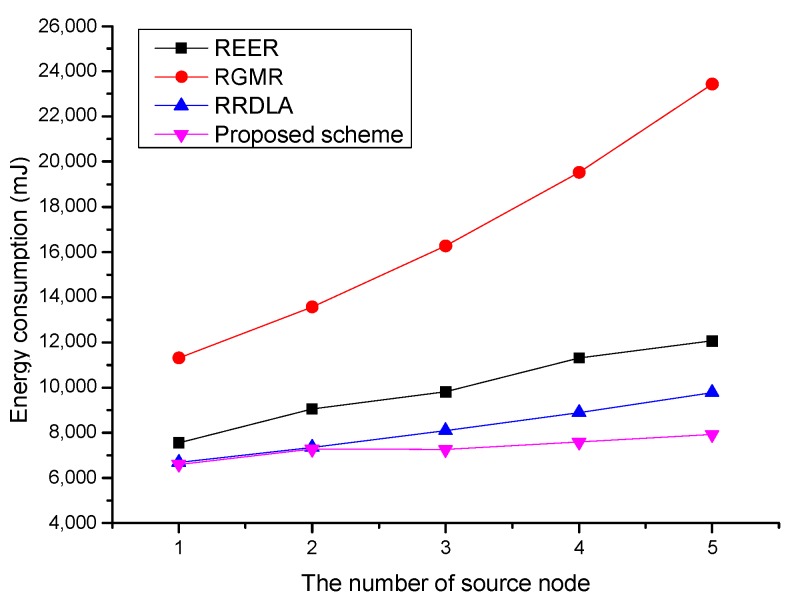
Average energy consumption vs. the number of source nodes with 90% of the link reliability.

**Figure 12 sensors-19-04072-f012:**
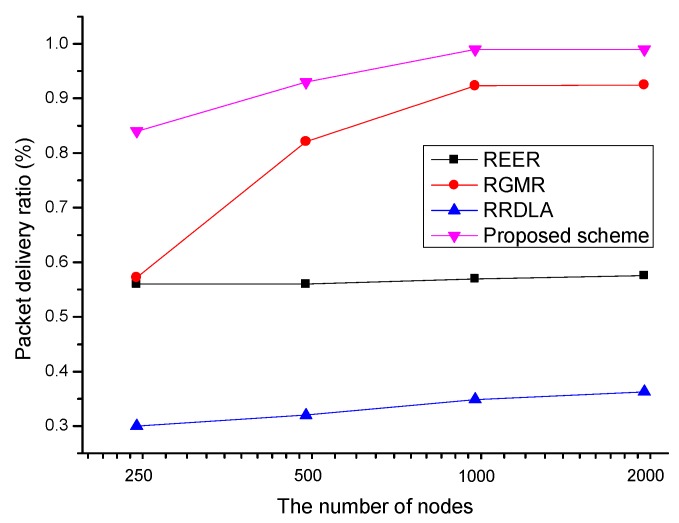
Packet delivery ratio vs. the number of nodes with 90% of the link reliability.

**Figure 13 sensors-19-04072-f013:**
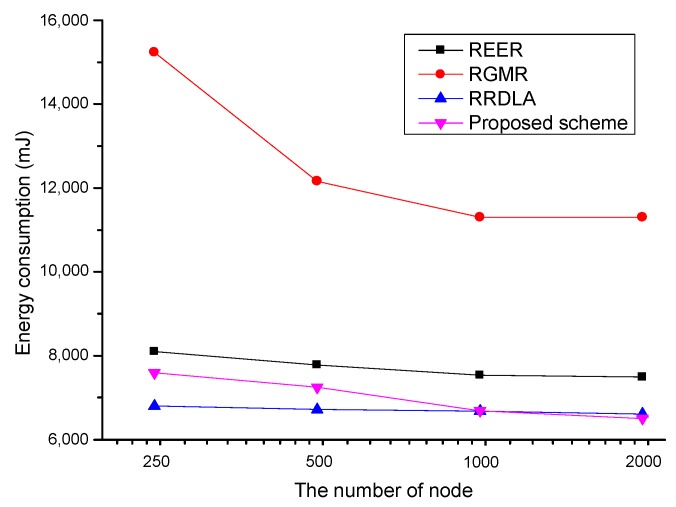
Average energy consumption vs. the number of nodes with 90% of the link reliability.

**Table 1 sensors-19-04072-t001:** Reliable routing protocols based on multipath routing, opportunistic routing, and a learning-based algorithm.

Category	Routing Protocols	Key Features	Reliability	Energy Consumption	Scalability (Long-Hop)
Multipath Routing	MMSPEED	High reliability and supporting real-time transmission	High	High	High
	REER	Reliability based on node-disjoint multipath	Medium	Medium	Low
	RGMR	High reliability based on radio-disjoint multipath	High	High	Medium-High
	RMER	Maximizing network life time based on load balancing	Low	Medium-High	Low
	ERMDT	Avoiding congestion area	Medium	Medium	Low
Opportunistic Routing	AOR	Preventing energy hole	Medium	Low-Medium	Low
	RTGOR	Supporting real-time transmission	Medium	Low-Medium	Low
Learning Algorithm	FTRP	Node logical clustering with learning phase for reliability	Medium-High	Low-Medium	Low
	RRDLA	Reliability based on learning algorithm and low energy consumption	Medium	Low	Low

**Table 2 sensors-19-04072-t002:** Time complexity of the proposed scheme and other protocols.

Category	Routing Protocol	Time Complexity
Multipath Routing	REER	O(N)
	RGMR	O(N×P)
Learning Algorithm	RRDLA	O(N×K)
Opportunistic Multipath Routing	Proposed Scheme	O(N×P) ≈ O(N)

**Table 3 sensors-19-04072-t003:** Simulation environment setting.

Parameter	Value (s)
Routing protocol	REER, RGMR, RRDLA, Proposed Scheme
Terrain	(1000 m, 1000 m)
End-to-end distance	800 m
The number of node	1000 Nodes
Node placement	Uniform & Random placement
Transmission range (m)	100 m
MAC protocol	802.15.4 MAC
MAC layer	CSMA/CA
Bandwidth	250 Kb/s
Payload size	32 bytes
Buffer size	24 (768 bytes)
Energy consumption (Tx)	24.92 mJ per 1 byte
Energy consumption (Rx)	19.72 mJ per 1 byte
Required packet delivery ratio	90%
Average link reliability	80% (75%∼85%), 90% (85%∼95%)

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
