# Peer review of "Opportunistic Multipath Routing in Long-Hop Wireless Sensor Networks"

_sensors, 2019, doi:10.3390/s19194072_

Round 1

Reviewer 1 Report

How to get related messages for link reliability, and which parameters are used to determine link reliability. Hope these can be expressed in the interpretation of the algorithm1. This paper mentions timer for waiting according to distance from j to destination in algorithm 2,and it will be better to explain it in further detail. For page typesetting, is it possible to write the algorithm into the corresponding section to make it easier to read and understand. This paper proposes an opportunistic multipath routing, but it is necessary to introduce and summarize its whole operation mechanism. This paper is insufficient in theory.

Author Response

(C1): How to get related messages for link reliability, and which parameters are used to determine link reliability. Hope these can be expressed in the interpretation of the algorithm1.

(R1): Thank you for the comments. We agree that it is necessary to explain a more detailed description of how to get a link reliability(quality) because the link reliability is one of important factors in our protocol. Thus, we have reflected this comment by p. 6, lines 194-200. The link reliability(quality) is calculated through the number of received beacon message over some time. If the transmission cycle of beacon message is 1 second and a node A received 7 beacon message during 10 seconds from a neighbor node B, the link quality between A and B is 70%.

(C2): This paper mentions timer for waiting according to distance from j to destination in algorithm 2, and it will be better to explain it in further detail.

(R2): We agree with your suggestion. Hence, we have incorporated your comments by p. 8, lines 257-267. The waiting time of a node A is calculated through the distance from A to destination, expected hop counts from A to destination and a constant k. If the distance is 500m and expected hop counts is 5 and k is 0, the waiting time could be 100ms. The k is exploited to modulate the time duration according to node specifications, network congestion, and other requirements.

(C3): For page typesetting, is it possible to write the algorithm into the corresponding section to make it easier to read and understand.

(R3): We agree with your opinion about readability. Therefore, we rearranged an algorithm, figure, and table to batch with their descriptions. Inevitably, if they and their description cannot be batched together on a page, we tried to batch them and their description as close as possible to make it easier to read and understand.

(C4): This paper proposes an opportunistic multipath routing, but it is necessary to introduce and summarize its whole operation mechanism. This paper is insufficient in theory.

(R4): To supplement the understand of whole operation mechanism, we have added a new Fig. 1 and a new subsection 3.1 (p. 5, lines 166-186). In the subsection, we explain the operation process briefly.

Reviewer 2 Report

The authors study different protocols for Wireless Sensor Networks that optimize performance when the network becomes scalable. They propose a methodology based on opportunistic routing according to the reliability requirement. This method reduces energy consumption and increases the probability of satisfactory delivery of packages to their destination.

This is an interesting approach. Below are my comments to the authors:

The authors can complement their study on performance metrics for the Wireless Sensor Network and protocol analysis, in the reference cited in:

Del-Valle-Soto, C., Mex-Perera, C., Orozco-Lugo, A., Lara, M., Galván-Tejada, G., & Olmedo, O. (2014). On the MAC/Network/Energy performance evaluation of wireless sensor networks: contrasting MPH, AODV, DSR and ZTR routing protocols. Sensors, 14(12), 22811-22847.

Table 1 is very complete, but they could establish more comparison protocols in the Learning Algorithms category, in addition to RRDLA. In Table 2, regarding the configuration parameters, are the packet loss per link taken into account? Is it the same for all links in the simulation? Which? Why subsections 4.3 and 4.4 relate these sets of metrics? Energy consumption influences several metrics, it would be interesting to see which impacts more and why. The number of references is very small. The research work is very interesting and complete, but it is not well supported by a reliable and thorough state of the art.

Author Response

(C1): The authors study different protocols for Wireless Sensor Networks that optimize performance when the network becomes scalable. They propose a methodology based on opportunistic routing according to the reliability requirement. This method reduces energy consumption and increases the probability of satisfactory delivery of packages to their destination. This is an interesting approach. Below are my comments to the authors:

(R1): Thank you for your interest in our work. We have revised the manuscript to reflect your opinion and suggestion. Please see our detailed responses below.

(C2): The authors can complement their study on performance metrics for the Wireless Sensor Network and protocol analysis, in the reference cited in:

Del-Valle-Soto, C., Mex-Perera, C., Orozco-Lugo, A., Lara, M., Galván-Tejada, G., & Olmedo, O. (2014). On the MAC/Network/Energy performance evaluation of wireless sensor networks: contrasting MPH, AODV, DSR and ZTR routing protocols. Sensors, 14(12), 22811-22847.

(R2): Thank you for the comments. We have supplemented the Table 3 (p. 12) referring to the reference paper you mentioned. In addition, the new performance evaluations for the added performance metric (the number of nodes) are simulated and discussed in subsection 4.6 (pp. 16-17, lines 480-503),

(C3): Table 1 is very complete, but they could establish more comparison protocols in the Learning Algorithms category, in addition to RRDLA.

(R3): We have supplemented the section 2 and Table 1 with explanations (p. 4, lines 130-140) of new reference based on learning algorithms, called FTRP[1]. In FTRP, the nodes create a logical cluster through the learning phase, and it improves packet delivery ratio through retransmission based on ACK message at the cluster head level.

[1] Moursy, I.; Elderini, M.; A. Ahmed, M. FTRP: A Fault Tolerant Reliable Protocol for Wireless Sensor Networks; September 10–14, 2017; SENSORCOMM 2017, The 11th International Conference on Sensor Technologies and Applications.

(C4): In Table 2, regarding the configuration parameters, are the packet loss per link taken into account? Is it the same for all links in the simulation? Which?

(R4): The link reliability is considered in the simulation. All links have random link reliability within a specific range. The average link reliability of 80% means that all links have a link reliability between 75% and 85%. Likewise, the average link reliability of 90% means that all links have a link reliability between 85% and 95%. As we recognize this point is not clearly mentioned, we have clarified the Table 3.

(C5): Why subsections 4.3 and 4.4 relate these sets of metrics?

(R5): The subsection 4.3 (subsection 4.4 in revised manuscript) related End-to-end distance and Average link reliability. As the end-to-end distance increase, more paths should be created to meet the packet delivery ratio demanded by the application. Thus, increasing end-to-end distances leads to increase the energy consumption. Similarly, the more paths be created to meet the packet delivery ratio when the link reliability is lower. It also leads to increase the energy consumption. In conclusion, the end-to-end distance and average link reliability directly impact on the energy consumption.

The subsection 4.4 (subsection 4.5 in revised manuscript) related The number of source nodes. As the number of source nodes increase, the number of constructed paths on the network increase. Thus, the probability that a node participates in multiple paths from a different source node would be increased, and this phenomenon leads to increases in energy consumption. In addition, the transmission delay owing to frequent collision avoidance caused by CSMA/CA causes buffer overflow, and some packet omissions lead to degradation of packet delivery ratio.

We have added the related performance metric associated with description to Table 3.

(C6): Energy consumption influences several metrics, it would be interesting to see which impacts more and why.

(R6): We agree with you. Thus, we perform more simulation about the number of nodes, and have added a new Fig. 13, Fig. 14, and a new subsection 4.6. The simulation results show that as the number of node decrease, the packet delivery ratio decrease and energy consumption increase. It is because the decreases of number of node means the decreases of node density. That is, as protocols cannot construct the sufficient number of paths in the low-density network, the packet delivery ratio is degraded. In addition, the energy consumption increase because the protocols perform the many numbers of retransmission to satisfy the required packet delivery ratio.

(C7): The number of references is very small. The research work is very interesting and complete, but it is not well supported by a reliable and thorough state of the art.

(R7): We have added a new reference included FTRP. In addition, we have added several related works in the revision process according to the results of the review.

Reviewer 3 Report

The paper deals with an interesting but quite well-known problem. I found the paper well written and organized.

To see the paper accepted the authors should address:

1) a discussion about the trade off with and without multipath: which the cost of having the multipath with respect to single path? This point is not clear in the current version

2) Energy issue could be solved with partial coverage too (multipath could be in contrast with this, so a "qualitative" discussion is needed. Consider [ref1, ref2, ref3] as reference)

3) A number of different approaches to multipath (in wired networks too) have been considered. And together to multipath at router level other multipath (overlay) can be considered. Discuss 

  4) Motivate the use of the topologies used in the simulation.

5) What about the performance and hw/sw capability of doing multipath at intermediate nodes? Are WSN nodes able to perform the proposed multipath algorithms?

[ref1] https://doi.org/10.1016/j.jnca.2016.12.022  [ref2] https://doi.org/10.1109/WCNC.2002.993520 [ref3] https://doi.org/10.1109/MILCOM.2001.985819

Author Response

(C1): The paper deals with an interesting but quite well-known problem. I found the paper well written and organized. To see the paper accepted the authors should address:

(R1): Thank you so much for your positive comments for our work. In the revised version we have made it clear for the questions. Please see our detailed responses below.

(C2): 1) a discussion about the trade off with and without multipath: which the cost of having the multipath with respect to single path? This point is not clear in the current version

(R2): Thank you for the comments. We have redrafted the section 1 to describe discussion about trade off with and without multipath in WSNs (p. 1, lines 15-23). The single path routing could discovery the route with minimum computational complexity and resource utilization, however, the single path is difficult to satisfy the packet delivery ratio owing to unreliability of wireless link in real world. On the other hand, the multipath routing could satisfy the packet delivery ratio through concurrently packet transmission over the multipath. Nevertheless, the multipath require more computational complexity and energy consumption to construct multipath. In conclusion, both single path and multipath routing protocol have their advantages and disadvantages.

(C3): 2) Energy issue could be solved with partial coverage too (multipath could be in contrast with this, so a "qualitative" discussion is needed. Consider [ref1, ref2, ref3] as reference)

(R3): The references discuss reducing energy consumption and prolonging the network lifetime. The [ref 1] solve the partial coverage problem based on learning automata to reduce a participation of the node in the packet transfer. As a result, the [ref 1] could find a minimum set of nodes to cover the demand portion with preserving the connectivity among the nodes. That is, as the number of participating nodes for monitoring is reduced, the energy consumption is reduced and the network lifetime is prolonged. We have written above-mentioned discussion in section 1 (p. 2, lines 47-54).

We think that [ref 2] and [ref 3] are more relevant to the multipath routing (even though they are exploited to reduce energy consumption). Thus, a discussion about [ref 2] and [ref 3] would be covered in conjunction with the response for comment 4 (C4).

(C4): 3) A number of different approaches to multipath (in wired networks too) have been considered. And together to multipath at router level other multipath (overlay) can be considered. Discuss

(R4): Many number of multipath routing protocols are proposed to achieve various objective such as load balancing, QoS supporting, reliability, and fault-tolerance. The [ref 2] constructs multipath between source and destination, and each path is assigned a probability of being chosen, depending on the energy metric to prevent network partition. That is, the [ref 2] construct multipath for packet transmission and exploit them evenly according to energy metrics. The [ref 3] proposed two options for prolonging network lifetime. The first option is to combine/fuse data generated by different sensors to minimize the energy consumption. The second option focuses on the paths. The [ref 3] exploit information on battery reserve and the energy cost to find the optimal routes. The options would be a guideline that aims at spreading the network traffic. The packet transmission according to options could reduce the overall energy consumption and distribute the traffic. As a result, the [ref 3] could achieve prolonging network lifetime.

We agree that an above-mentioned discussion is important point, however, we believe that the studies would be slightly outside the scope of our paper because we mainly focus on improvement packet delivery ratio. Thus, we briefly summarized and written the discussion in section 1 (p. 2, lines 24-27)

(C5): 4) Motivate the use of the topologies used in the simulation.

(R5): We have incorporated your comments. We have added a new Fig. 5 (p. 13) to further illustrate our simulation topology and explained more details about effect of the environment by p. 12, lines 377-380.

(C6): 5) What about the performance and hw/sw capability of doing multipath at intermediate nodes? Are WSN nodes able to perform the proposed multipath algorithms?

(R6): To verify the proposed scheme could be operated on WSN nodes, we analysis the time complexity. We have included a new Table. 2 (p. 11) and a new sub chapter 4.1 (p. 12, lines 355-373) to describe the analyzed time complexity. Through the time complexity analysis, we confirmed that the proposed scheme has a similar time complexity with the existing studies. Thus, we think that the WSN nodes could be able to perform the proposed multipath routing protocol.

[ref1] https://doi.org/10.1016/j.jnca.2016.12.022 

[ref2] https://doi.org/10.1109/WCNC.2002.993520

[ref3] https://doi.org/10.1109/MILCOM.2001.985819

Round 2

Reviewer 1 Report

In section 3.1, it is better to introduce and summarize its whole operation mechanism roundly, including not only the process of routing establishment, but also the method of obtaining beacon message, and so on. Give a brief explanation of how to determine the value of T, TB and H(j,Dest). In addition, the more messages a node receives and forwarded, the sooner it dies because of too much energy consumption, so is it possible to explain the reasonableness of the definition of Pj.

    3.This article is duplicated in lines 168 to 170. Please check carefully.

Author Response

(C1): In section 3.1, it is better to introduce and summarize its whole operation mechanism roundly, including not only the process of routing establishment, but also the method of obtaining beacon message, and so on.

(R1): Thank you for the comments. We have supplemented the subsection 3.1 to describe the preconditions and behavior of the nodes like the exchanging of beacon messages. (p.5, lines 166 -171)

(C2): Give a brief explanation of how to determine the value of T, TB and H(j,Dest).

(R2): We have added a brief explanation of how to get the values. T and T_B have explained in subsection 3.1(p. 5, line 167-168 and p.5, 170-171, respectively). The H(j,Dest) has explained in subsection 3.3(p. 9, line 266).

(C3): In addition, the more messages a node receives and forwarded, the sooner it dies because of too much energy consumption, so is it possible to explain the reasonableness of the definition of Pj.

(R3): We agree that if a node is exploited repeatedly, the lifetime of the node would be shorter than other nodes.

This problem is usually mentioned in the protocol which is to exploit the designated route. The existing multipath routing protocols including REER and learning based routing protocols like RRDLA would face the problem because they transfer a packet over the same path(nodes).

On the contrary, the proposed scheme that gives all neighbors the opportunity to transmit a packet through opportunistic routing could expect a relatively long lifetime of the nodes (even though the nodes with higher link reliability are exploited more often).

Finally, as the problem is directly related to the network lifetime, it is one of the most important factors in WSNs. Thus, we are currently performing the study to mitigate the concentrated use of some nodes by load shifting concept.

(C4): 3.This article is duplicated in lines 168 to 170. Please check carefully.

(R4): We have checked and corrected the error.

Reviewer 3 Report

accept in the current form

Author Response

(C1): accept in the current form

(R1): Thank you for giving us the opportunity to strengthen our manuscript with your valuable comments and queries.